# Mitigation of Muscle Loss in Stressed Physiology: Military Relevance

**DOI:** 10.3390/nu11081703

**Published:** 2019-07-24

**Authors:** David D. Church, Jess A. Gwin, Robert R. Wolfe, Stefan M. Pasiakos, Arny A. Ferrando

**Affiliations:** 1Department of Geriatrics, Donald W. Reynolds Institute on Aging, Center for Translational Research in Aging & Longevity, University of Arkansas for Medical Sciences, Little Rock, AR 72205, USA; 2Military Nutrition Division, U.S. Army Research Institute of Environmental Medicine, Natick, MA 01760, USA; 3Oak Ridge Institute for Science and Education Supporting the Military Nutrition Division of the US Army Research Institute of Environmental Medicine, Natick, MA 01760, USA

**Keywords:** skeletal muscle, military, protein metabolism, amino acids, testosterone, androgens

## Abstract

Military personnel may be exposed to circumstances (e.g., large energy deficits, sleep deprivation, cognitive demands, and environmental extremes) of external stressors during training and combat operations (i.e., operational stressors) that combine to degrade muscle protein. The loss of muscle protein is further exacerbated by frequent periods of severe energy deficit. Exposure to these factors results in a hypogonadal state that may contribute to observed decrements in muscle mass. In this review, lessons learned from studying severe clinical stressed states and the interventions designed to mitigate the loss of muscle protein are discussed in the context of military operational stress. For example, restoration of the anabolic hormonal status (e.g., testosterone, insulin, and growth hormone) in stressed physiological states may be necessary to restore the anabolic influence derived from dietary protein on muscle. Based on our clinical experiences, restoration of the normal testosterone status during sustained periods of operational stress may be advantageous. We demonstrated that in severe burn patients, pharmacologic normalization of the anabolic hormonal status restores the anabolic stimulatory effect of nutrition on muscle by improving the protein synthetic efficiency and limiting amino acid loss from skeletal muscle. Furthermore, an optimal protein intake, and in particular essential amino acid delivery, may be an integral ingredient in a restored anabolic response during the stress state. Interventions which improve the muscle net protein balance may positively impact soldier performance in trying conditions.

## 1. Introduction

The skeletal muscle system is most often a topic of discussion in relation to physical function. In addition, muscle serves as the body’s primary protein reserve. Muscle mass and strength are crucial for health, physical function, and athletic performance in all populations [1]. In healthy trained athletes, increased muscle strength and mass are often important goals for achieving a maximal performance. With increasing physiological stress, the maintenance of muscle mass becomes a primary determinant of both quality of life and survival [2]. In stressed physiological states, muscle serves as a metabolic reserve by providing essential amino acids (EAA) for gluconeogenesis, as well as splanchnic, immunological, and wound-healing protein synthesis requirements [3,4]. In a clinical setting, muscle represents a key metabolic component required to combat the prevailing pathology. These roles of muscle are also relevant to circumstances such as multi-stressor military operations, in which health physiology is significantly challenged. Contrary to this clinical scenario, there are certain circumstances, such as multi-stressor military operations, where health physiology is significantly challenged. This situation provides a unique opportunity to study the effects of severe stress on muscle mass in otherwise healthy individuals. For military personnel, understanding the physiological consequences of severe stress on muscle mass is critical for the design, assessment, and implementation of pharmacological and/or nutritional strategies to conserve muscle mass and, most importantly, function.

Exposure to catabolic states is common in some military personnel, including light infantry and special operations forces (SOF), who, at times, may be exposed to a wide range of diverse stressors during sustained training and combat operations (SUSOPS). For example, energy deficits of 2500–4000 kcal·day^−1^ have been reported during operational situations [5,6,7,8]. Furthermore, Norwegian men undergoing 7 days of field training lost an average of 4 kg of fat-free mass [9]. The loss of body protein during simulated combat operations is the result of altered protein kinetics that favor catabolism [10,11,12], which in turn may compromise performance [13]. We will use our previous investigations of critical care and severe stress to discuss potential mechanisms and strategies to combat muscle loss in military personnel during SUSOPS.

## 2. Lessons Learned from Clinical Stress States

Various clinical stress states have been studied and documented in the literature [14]. One of the most severe is the hypermetabolic and catabolic state that occurs in response to severe burn injury. Burn injury induces anabolic resistance in muscle, as muscle transitions from metabolic maintenance to catabolism and the provision of constituent amino acids (AA) for higher priority tissues. This alteration in muscle metabolism can persist for up to a year after the injury has fully healed [15]. Burn patients display some of the largest negative net protein balances (NB) reported in the literature [14,16], and although the catabolic state is a response to an injury, interventions that can ameliorate the prevailing catabolism may be relevant to stressed states in healthy individuals. For example, certain physiological perturbations are present in both burn patients and military personnel, such as elevated metabolic rates and altered hormonal secretion and/or influence. In burn patients, the rate of muscle protein breakdown (MPB) exceeds that of muscle protein synthesis (MPS), leading to a catabolic state that, if not corrected, results in large losses of skeletal muscle mass [4]. Even though MPS is elevated in these patients, the greater increase in MPB indicates that the majority of muscle protein is being broken down to supply the higher priority areas of the body with AA [3]. The only means of reducing the dramatic loss of muscle protein, and its constituent AA, with burn injury is to restore the ability of muscle to respond to anabolic stimuli. In addition, the stressed states generally suppress anabolic hormone production and/or elicit an anabolic resistance to those hormones. Overall, the greater the stressed physiology, the greater the propensity for the loss of muscle protein.

## 3. Role of Muscle in the Stressed State

The fundamental mechanism mediating changes in muscle mass is the NB between MPS and MPB (i.e., NB = MPS − MPB). A net gain in muscle protein occurs when NB is positive, indicating an anabolic state. A net loss, or catabolic state, occurs when MPB exceeds MPS. In the catabolic state, there is a significant acceleration of protein breakdown relative to the rate of synthesis in the post-absorptive state in both skeletal muscle and the whole body [14]. AA cannot be “stored” in the body in the same manner as lipids and carbohydrates. Incorporation of AA into proteins, enzyme function in metabolic pathways, conversion of their carbon skeleton to fatty acids, and excretion as urea or ammonia serve as the possible fates of AA. The lack of AA storage dictates that muscle protein is the sole source of EAA for the rest of the body. Once AA concentrations decrease in response to prolonged starvation/energy deficit, the MPB rate is increased in order to maintain blood AA concentrations until food or protein is provided. This mechanism is effective at maintaining blood AA concentrations until body protein is exhausted, at which point the individual dies [1].

In the absence of the dietary intake of protein, muscle protein serves as the body’s only source of precursor AA, particularly the EAA. As the cumulative stress on an individual increases in the absence of food, muscle protein NB becomes more negative. With increasing stress, the metabolic priority of muscle is to deliver AA to higher priority functions necessary to facilitate survival. These include the support of acute-phase protein synthesis, immune function, wound healing, energy production, etc. [3,17]. During periods of high stress, the maintenance of muscle mass and the continued supply of EAA become essential for health and survival. For example, obese humans and animals die after the depletion of body protein when subjected to prolonged starvation, even when energy stores have not been depleted [18,19]. During starvation, the body’s main metabolic focus is the maintenance of body protein. In fact, a small amount (7.5 g) of carbohydrate intake can reduce urinary nitrogen loss by ~40%–50% [20]. Notably, this effect is not further enhanced with the provision of 15–25 g or 54–108 g of carbohydrate [17,20]. Taken together, these data indicate that muscle protein is likely the main determinate of long-term survival in humans, particularly in response to stress [2].

The magnitude of stress affects both the direction and magnitude of NB. For example, resistance exercise increases protein turnover and results in a less negative NB in healthy, normal humans [21]. Oral protein and AA ingestion following an acute bout of resistance exercise creates positive NB [22]. However, with greater physiological stress comes a greater attenuation of the anabolic stimulus to ingested protein/AA, even when given in abundance [23]. When the physiological stress results in skeletal muscle anabolic resistance, nutrition alone is generally not able to match the body’s requirement for AA. In this regard, elevated rates of MPB are required to provide the needed precursor AA for splanchnic (acute-phase), immunological, and wound healing protein synthesis (Figure 1) [4]. While this counter-regulatory response of muscle protein is beneficial to survival in the short term, a prolonged catabolism of muscle protein leads to continued muscle mass loss. The challenge is to develop strategies that counter this physiological phenomenon.

## 4. Amelioration of Muscle Loss in Clinical Stress

In a stressed state such as critical illness, burn, severe injury, and major surgery, the energy requirement to accommodate the resulting hypermetabolism is reasonable. In studies in severely burned (growing) children, measurement of the total energy expenditure with doubly-labelled water indicated only an 18% increase above resting energy expenditure [24]. The delivery of a larger number of kilocalories to patients via parenteral and enteral nutrition does little to ameliorate the loss of muscle mass. In fact, it can be counterproductive and, perhaps, harmful, as the delivery of large amounts of energy can contribute to lipid accumulation in the liver [25]. This suggests other factors which accompany the hypermetabolic response are responsible for the loss in muscle mass, namely the increased whole-body requirement for AA, which are derived from muscle (Figure 1). The body’s AA requirements for acute phase protein synthesis and the synthesis of proteins related to immune function and wound healing are so great that these patients are in a catabolic state despite constant enteral feeding [23]. In fact, MPB in a postprandial burn patient is more than twice that of a fasted, healthy subject [16]. Muscle protein turnover is influenced by hormonal stimuli in a healthy person, a process which serves to rebuild and replace proteins to preserve muscle function and protein homeostasis. During periods of substantial physiological stress, the accompanying neuroendocrine changes are directed towards survival. The hormonal environment in burn patients is characterized by marked hypercortisolemia and hypoandrogenemia [23]. The combination of prolonged elevations in cortisol and the absence of the anabolic influence from testosterone, growth hormone, and insulin, serves as a persistent signal for MPB. The persistent negative NB not only occurs in the post-absorptive state, but also in response to exogenous AA [23]. These physiological alterations are not unique to burn injury, as other clinical stressors, such as sepsis, lead to an increased rate of whole-body protein catabolism, regardless of the nutritional support status [26]. Therefore, despite the presence of an abundance of AAs in circulation, their inward transport into skeletal muscle is compromised and not sufficient to offset the negative NB across muscle. In this circumstance, it may be difficult to restore NB without restoration of the normal anabolic hormonal influence. A variety of anabolic agents have been investigated in burn patients, and it has been demonstrated that the administration and/or normalization of testosterone and/or its oral analogue, oxandrolone, growth hormone (GH), insulin, insulin-like growth factor (IGF)-1, and IGF-1 plus IGF-1 binding protein-3 (IGFBP3), will all improve muscle NB and mitigate the efflux of muscle amino acids [4]. Not surprisingly, these hormones have been shown to be altered by SUSOPS and military training [5,6,10,27], indicating a systemic amelioration of anabolic influence on skeletal muscle with increasing physiological stress. Despite the clinical success of the restoration and/or normalization of various hormonal effects, testosterone administration remains the most pragmatic, safe, and logistically feasible paradigm for consideration in military personnel.

## 5. Applications to Military Personnel

Studies in military personnel conducting multi-stressor SUSOSP (Figure 2) provide a unique opportunity to investigate an otherwise healthy, non-pathological physiology subjected to unavoidable stressors that disrupt metabolism and muscle and whole-body protein homeostasis. These stressors include prolonged periods of strenuous physical work, limited access to food in hot and cold conditions, and exposure to a high altitude (i.e., hypobaric hypoxia).

Independent of the environmental, physical, and nutrition-related stress, military personnel are exposed to other stressors, including sleep deprivation and significant cognitive demands that may exacerbate the consequences of any of the perturbative stressors inherent to SUSOPS. For example, one study involved a simulated reconnaissance and support mission which required SOF to stealthily infiltrate a wooded area to observe a potential weapon sale [28]. The mission required soldiers to remain undetected in a prone position for an 8-day period, during which a high degree of mission stress was maintained (presence of guard dogs and helicopters). Soldiers could only move when crawling supine to an observation point in close proximity to the resting point. The soldiers were provided with 3 liters of water and standardized field rations (3567 kcal, 84g protein, 547g carbohydrates, and 116g fat) each day [29]. Despite the provision of what appeared to be adequate food, lean mass decreased by 5%. The loss of lean mass was accompanied by a reduced performance, as the training operation produced an 8.2% decrease in the vertical jump height, a 9.2% decrease in the maximal force, a 21.9% decrease in the rate of force development, and alterations in the muscle fiber type [28,29]. Energy intake was not tracked, so an energy deficit, which alone can attenuate MPS [30], was not ruled out; however, these studies highlight that even without high energy expenditure and physical activity levels, military personnel can experience decrements in muscle mass, morphology, and function.

## 6. Military Operational Energy Deficits

Combat rations are designed to meet the average nutritional requirements during most operational scenarios. Energy expenditure, measured via doubly labelled water, from SOF personnel during 12 different SOF training scenarios ranged from 3700 to 6300 kcal·day^−1^ (average of ~4500 kcal·day^−1^). These data indicate that even if the provided rations were consumed in their entirety, SOF personnel would still be in an energy deficit. Furthermore, the SOF courses that produced the highest energy expenditures also had the lowest energy intakes, and therefore, the largest energy deficits [7]. This is likely because missions with higher physical activity demands provide not only less time to eat, but less motivation to do so. Field operations place a high premium on activities that compete with eating. Tactical operations dictate that military personnel choose the mission over eating. In the US Air Force Survival Course, students consumed only 60% of the rations provided [31]. More recently, Margolis [11] demonstrated that soldiers will consume ration items containing extra protein or carbohydrate, but will compensate for the extra energy by eating less of the standard ration components. In the same study, soldiers who were able to consume enough of their rations to minimize their energy deficits to ~39% displayed a positive whole-body NB, though only 20.5% of participants were able to do so. In this regard, it is important to note that an increased protein intake is necessary to help retain body nitrogen during energy deficit [32]. Military personnel also traditionally carry large loads in the field, and this has only increased over time [33]. This often includes rations and occupational items required to carry out the mission. However, given the choice of items to carry, military personnel may be more tempted to trade food for necessities such as dry clothing [34]. As a result, military personnel often deliberately choose to “field-strip” their rations for a mission, thereby sacrificing their energy intake to improve the pack weight or to maintain essential personal or operational items [34]. Therefore, the issue of energy deficit during military operations is not only an issue of energy delivery to meet elevated energy demands, but also one of energy consumption. While the majority of these studies have been short-term, a further exacerbation of outcomes would be expected with prolonged energy deficient states.

## 7. Strategies to Combat Energy Deficient States

Given the prevalence of energy deficits in military personnel during SUSOPS, several strategies have been evaluated for their efficacy to reduce the resultant loss of muscle mass. Increasing dietary protein intake to double and triple the Recommended Dietary Allowance (RDA, 1.6 and 2.4 g protein·kg^−1^·d^−1^) has been shown to maintain the anabolic response to feeding and preserve lean mass during 21 days of moderate (40%) energy deficit when compared with protein intakes at the RDA (0.8 g protein·kg^−1^·d^−1^) [35]. Protein supplementation following 2 weeks of a 28–34% energy deficit was able to restore MPS to rates observed during energy balance [36], as energy deficits normally attenuate MPS [30]. The muscle protein sparing effects of higher-protein diets during energy deficient states are enhanced with the addition of resistance-type exercise. The addition of resistance exercise and high-intensity interval training to 2.4 g·kg^−1^·d^−1^ protein intake during 4 weeks of 40% energy deficit resulted in an actual gain in lean mass in healthy young men [37]. The increase in lean mass appears to be attributed to resistance exercise, as consuming the same amount of protein in another study only attenuated lean mass loss when the exercise intervention that was primarily performed was aerobic in nature [35]. The combined stimulatory effect of dietary protein and resistance exercise during energy deficit has been demonstrated in a variety of populations [38]. Taken together, resistance exercise and high protein intakes help restore and provide the necessary anabolic stimulus to spare muscle mass during energy deficits of up to approximately 40% of the total energy requirements.

These studies have been useful; however, they assess a single stressor (energy deficits) in a controlled laboratory setting. These studies cannot replicate the influence of constant mental and physical stress during real-world military operations. The dilemma of supplying soldiers with adequate and appropriate nutrition has been recognized by military leaders for centuries [39]. The efficacy for maintaining the anabolic stimulus to muscle during a period of energy deficit appears to have a limit. When energy deficits are above 40%, high protein intakes do not appear to restore muscle anabolism to rates observed during energy balance. In this regard, AA become an oxidizable energy source to meet the energy demand [38]. This is a paramount issue for military personnel exposed to severe energy deficits (> 40% of total energy requirements) during SUSOPS. The MRE is designed to meet energy and nutrient requirements established by Army Regulation 40–25 if consumed as directed (i.e., 3 MRE per day, ~3900 kcal, 127 g protein, 507 g carbohydrate, 152 g of fat; [40,41]). However, despite the ration’s design and intended use, energy deficits and a suboptimal nutrient intake, as well as their associated consequences on muscle mass and whole-body protein, are largely inevitable during strenuous SUSOPS [10,11,12,39,42]. To date, several strategies have been employed to prevent energy deficits during military operations, including the provision of supplemental carbohydrate or protein [11], increasing the size and energy content of certain ration components (unpublished data), and pre-operational nutrition education [8]. A recent study was conducted during a 70% energy deficit while at altitude. One group received a high-protein diet (2.0 g protein·kg^−1^·d^−1^), while the other was on a standard protein diet (1.0 g protein·kg^−1^·d^−1^), for 21 days at 4300 m. The high-protein diet group displayed a more negative post-absorptive whole body protein net balance as almost all (0.95 ± 0.32 g protein·kg^−1^·d^−1^ more than the standard protein group) the additional protein was oxidized for energy [10]. In addition, the muscle was resistant to protein (25 g of whey protein) after aerobic exercise, as indicated by the inhibition of mammalian target or rapamycin complex 1-mediated signaling [43]. Therefore, dietary strategies to this point have not been successful. However, the lack of an apparent benefit gained by simply consuming more protein makes sense; borrowing from our clinical experience, as physiological stress increases, nutrition alone is not capable of sparing muscle mass.

For this reason, efforts have focused on optimal EAA delivery, rather than the consumption of a defined amount of protein. We have demonstrated that the EAA are the prime drivers of muscle anabolism [44,45]. Furthermore, we have also demonstrated that in anabolic resistance states, such as aging [46,47] and post-operative joint arthroplasty [48,49], adjusting the amount and/or individual ratios [47] of EAA intake can restore the anabolic response in skeletal muscle. While our current efforts are focused on optimizing EAA delivery, this aspect is part of an overall strategy to restore both the anabolic signal and the anabolic capacity of skeletal muscle in the stressed state. Little is known about the EAA requirement and optimal delivery format required to overcome the large hypocaloric state experienced by military personnel during field and combat situations. Whey protein [10,43,50] has been utilized in the past in hopes of ameliorating body protein losses; however, the energy deficit was such that protein/amino acid intake was directed towards energy production, rather than protein turnover. For this reason, current investigations are focused on determining the optimal EAA requirement for skeletal muscle and whole body protein during energy deficit.

## 8. Restoration of Hormonal Influence

The inability of nutrition to preserve muscle mass in military personnel on its own is similar to the anabolic resistance displayed in severe burn patients receiving enteral nutrition [16,51]. Like burn patients, military personnel demonstrate an inhibition of endogenous testosterone production and a demonstrated hypercortisolemia [5,6,10,27], though to a lesser degree. Interestingly, the restoration of testosterone concentrations [4] has improved NB in severe burn patients. Testosterone has been shown to maintain muscle mass through two primary mechanisms: 1) reinstating the anabolic response of muscle to AA and 2) improving the protein synthetic efficiency in the post-absorptive state [23,52,53,54,55,56]. As previously mentioned, MPS in the burn patient does not respond to feeding [16,51]. However, the normalization of testosterone concentrations in adults and the use of oxandrolone in children restored the anabolic effects of enteral nutrition [4]. The return of the anabolic stimulus from feeding has also been demonstrated with insulin, IGF-1, and GH, where the stimulation of MPS is closely related to an increased rate of inward AA transport [4,57]. Therefore, the systematic administration of these hormones can only stimulate MPS if AA are present [56]. In contrast, mitigation of the hypogonadal state with exogenous testosterone has the advantage of increasing the reutilization of intracellular AA, thereby improving the protein synthesis efficiency [23,52,55,56]. In other words, for a given quantity of muscle intracellular AA (particularly the EAA), testosterone normalization directs them toward the synthetic process, rather than outward transport for utilization elsewhere. This has been shown to result in an improvement in muscle NB [54,55]. This mechanism of action may be advantageous for military personnel often exposed to conditions that elicit a severe energy deficit and hypogonadism [5,58]. It is plausible that the restoration of a eugonadal state would route MPB-derived AA back into muscle protein (via improved synthetic efficiency), rather than releasing them into the blood. Therefore, hormonal normalization provides a potential pharmacological strategy for military personnel to attenuate muscle mass and, importantly, functional declines. Combined with an optimal nutrition strategy, restoration of the anabolic signal may restore the effects of EAA on skeletal muscle.

## 9. Preliminary Investigations Utilizing Testosterone

Efforts are currently underway to investigate the utilization of testosterone in military personnel exposed to strenuous SUSOPS [59]. We suspect that given their habitual energy deficient and hypogonadal state during SUSOPS, testosterone administration, based upon our clinical findings, would restore the anabolic response derived from protein/AA intake. In particular, we would hypothesize that the improved synthetic efficiency in muscle will mitigate the loss of AA in the fasted state. Additionally, the restoration of anabolic influence by returning to a eugonadal state will restore the anabolic effects of protein/AA on skeletal muscle. Efforts involve both the administration of testosterone enanthate every 7 days, as well as the use of other, long-term testosterone esters that can maintain testosterone concentrations within the normal physiological range for 8–10 weeks [60]. Both administration paradigms may have logistical and physiological merit for use in the context of multi-stressor SUSOPS.

## 10. Summary

Although the present review focused on the responses in men, the number of women involved in combat roles has continued to increase. At present, there is a dearth of work delineating the effects of military operations on women. Future research is needed to determine the physiological and metabolic responses of women to the cumulative stress of SUSPOS, and to what extent these responses differ from men. Nonetheless, current evidence indicates that military personnel are, at times, exposed to multi-stressor environments for sustained periods that can decrease muscle and whole-body protein [5,6,9,10,11,12,33,59]. Borrowing from clinical stress examples, greater physiological stress is associated with an absence of anabolic influence on muscle, and further, the resistance of muscle to the effects of nutrition/protein. In order to restore the anabolic influence of AA, hormonal signaling must be restored and/or enhanced, potentially providing military personnel with a feasible strategy to maintain muscle mass and function during SUSOPS. While the restoration or enhancement of most hormonal signals in burn patients improves muscle NB with nutrition, testosterone administration is particularly noteworthy due to its economic, logistic, and metabolic advantages. In this regard, we propose the exploration of potential pharmacological solutions that would restore the anabolic potential in muscle and, when combined with a targeted nutritional strategy, mitigate the loss of muscle protein and function in military personnel in response to multi-stressor SUSOPS.

## Figures and Tables

**Figure 1 nutrients-11-01703-f001:**
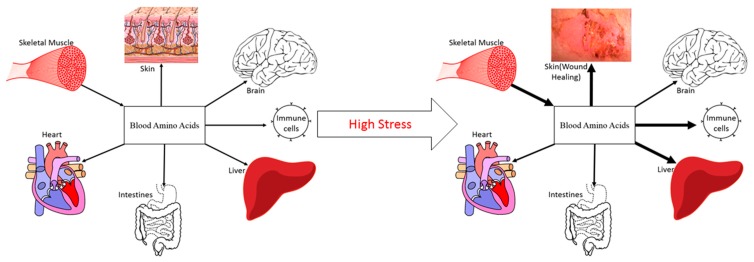
High stress induces muscle catabolism, resulting in the release of amino acids into circulation to be used for higher priority survival processes (representative example; not an exhaustive list; not drawn to scale).

**Figure 2 nutrients-11-01703-f002:**
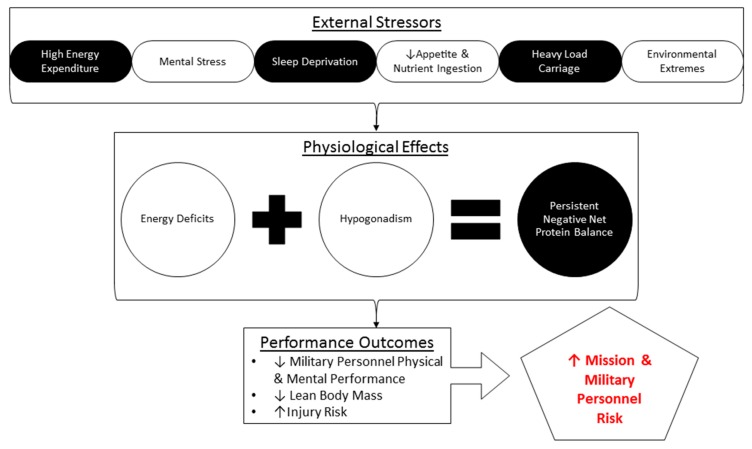
Potential stressors that military personnel may be exposed to during strenuous sustained training and combat operations (SUSOPS).

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
