# Peer review of "Mitigation of Muscle Loss in Stressed Physiology: Military Relevance"

_nutrients, 2019, doi:10.3390/nu11081703_

Round 1
Reviewer 1 Report
This manuscript review the mitigation of muscle loss in military personnel exposed to external stressors during training and combating operations, including prolonged periods of strenuous physical work, limited access to food in hot or cold conditions, and exposure to high altitude, by applying the clinical strategies.
The failure of ameliorating the muscle loss through the nutritional interventions including combating energy deficit , high protein intake and AA delivery suggests that the EAA are the prime drivers of muscle anabolism. Studies regarding the optimal EAA delivery need to be emphasized in military personnel.
Restoration of hormonal influence in burn patients was applied in military personnel. However, information regarding the inhibition/low of endogenous testosterone production and hypercortisolemia in military personnel is limited? [ Ref. 10 demonstrates that consuming high protein diet did not protect fat free mass during severe energy deficit at high altitude. Ref. 23 Testosterone administration in severe "burn patients" ameliorates muscle catabolism.]
From clinical experience (burn patients), nutrition alone is not capable of sparing muscle mass as physiological stress increases. Hence, administration of exogenous testosterone to increase the intracellular AA (particularly EAA), thereby improving protein synthesis efficiency was suggested. However, The return of anabolic stimulus from feeding has also been demonstrated with insulin, IGF-1 and GH. How do these hormones influence the muscle mass in military personnel?
Author Response
Reviewer: This manuscript review the mitigation of muscle loss in military personnel exposed to external stressors during training and combating operations, including prolonged periods of strenuous physical work, limited access to food in hot or cold conditions, and exposure to high altitude, by applying the clinical strategies.
Reviewer: The failure of ameliorating the muscle loss through the nutritional interventions including combating energy deficit, high protein intake and AA delivery suggests that the EAA are the prime drivers of muscle anabolism. Studies regarding the optimal EAA delivery need to be emphasized in military personnel.
Authors: We appreciate and agree with the reviewer, however, little is known about the optimal EA delivery in military personnel. For this reason, our current efforts are focusing on EAA optimization. We have highlighted this fact and included what work has been done; lines 256 – 262.
Reviewer: Restoration of hormonal influence in burn patients was applied in military personnel. However, information regarding the inhibition/low of endogenous testosterone production and hypercortisolemia in military personnel is limited? [ Ref. 10 demonstrates that consuming high protein diet did not protect fat free mass during severe energy deficit at high altitude. Ref. 23 Testosterone administration in severe "burn patients" ameliorates muscle catabolism.]
Authors: Thank you for pointing out this issue, we know cite multiple studies that demonstrate low of endogenous testosterone production and hypercortisolemia in military personnel.
Reviewer: From clinical experience (burn patients), nutrition alone is not capable of sparing muscle mass as physiological stress increases. Hence, administration of exogenous testosterone to increase the intracellular AA (particularly EAA), thereby improving protein synthesis efficiency was suggested. However, The return of anabolic stimulus from feeding has also been demonstrated with insulin, IGF-1 and GH. How do these hormones influence the muscle mass in military personnel?
Authors: This is a good point and in short, all the anabolic hormones are lowered as a result of SUSOPS. As such it would be hard to identify the exact roles all of these hormones have individually. We have included information addressing this; lines 149 – 153.
Reviewer 2 Report
This is a succinct yet comprehensive review of the physiology of muscle protein balance in stressed states and its application to intense military training and operational situations. It provides important evidence for the limitations of diet and protein (EAA) intake in completely mitigating loss muscle mass and function and suggests the importance and relevance of hormonal regulation and supplementation in possible mitigating interventions in these situations. The one piece perhaps missing from the summary are considerations for sex differences as the review focused primarily on male responses. With females increasingly involved in military field operations an acknowledgement of the need for further research into possible sex differences in responses and interventions should be included in the review.
Author Response
Reviewer: This is a succinct yet comprehensive review of the physiology of muscle protein balance in stressed states and its application to intense military training and operational situations. It provides important evidence for the limitations of diet and protein (EAA) intake in completely mitigating loss muscle mass and function and suggests the importance and relevance of hormonal regulation and supplementation in possible mitigating interventions in these situations. The one piece perhaps missing from the summary are considerations for sex differences as the review focused primarily on male responses. With females increasingly involved in military field operations an acknowledgement of the need for further research into possible sex differences in responses and interventions should be included in the review.
Authors: We thank the reviewer for their comments, and agree that sex difference are an important consideration that needs to be addressed with military field trials. We have added (lines 302 – 305) this in the summary as a recommendation for future research studies.
Reviewer 3 Report
Summary: This is an interesting topic that deserves increased attention and further research to improve outcomes during physiologically stressing military operations. While there are many examples of clinical stress mentioned in line 120, burns are the only condition expanded upon throughout the review. It would be helpful to expand on more states than just burns to provide wider context and information on what has been investigated and found in other stressed states.
Specific comments:
1) In lines 242-248, the authors state that provision of EAA in anabolic resistant states such as aging can restore anabolic response in skeletal muscle. However, aging is not necessarily a stressed state as the others listed in line 120. Is there any evidence in burns or other clinically stressed states that altering the amount and/or ratios can restore the anabolic response to protein?
2) Line 272-273 is too presumptuous based on available evidence - change should to may
3) Line 288 - change to deplete to more accurate word... decrease, impact, diminish, start to deplete, etc.
4) Line 291-292: Seems inaccurate to say that muscle MUST be presented with an optimal pool of EAA precursors to restore the anabolic influence of AA. Based on burns research, the EAAs are not adequate to restore the anabolic response. Therefore, it is really the hormonal amounts/signaling that must be restored and/or enhanced. This alone will (in theory) restore the anabolic effect of protein. I didn't see any evidence provided that EAAs will overcome the anabolic resistance as stated in Comment 1. If such evidence exists, please provide.
Author Response
Reviewer: Summary: This is an interesting topic that deserves increased attention and further research to improve outcomes during physiologically stressing military operations. While there are many examples of clinical stress mentioned in line 120, burns are the only condition expanded upon throughout the review. It would be helpful to expand on more states than just burns to provide wider context and information on what has been investigated and found in other stressed states.
Authors: We agree with the reviewers point, and have highlighted sepsis and post-operative joint arthroplasty as additional clinical stresses in various parts of the manuscript. Lines: 139 – 141 and 253.
Specific comments:
Reviewer: 1) In lines 242-248, the authors state that provision of EAA in anabolic resistant states such as aging can restore anabolic response in skeletal muscle. However, aging is not necessarily a stressed state as the others listed in line 120. Is there any evidence in burns or other clinically stressed states that altering the amount and/or ratios can restore the anabolic response to protein?
Authors: In burns feeding alone is not enough to restore the response. However, we have added that increased EAA intake intake in post-operative joint arthroplasty patients improves outcomes. Line: 253
Reviewer: 2) Line 272-273 is too presumptuous based on available evidence - change should to may
Authors: Agreed. We have made the appropriate change.
Reviewer: 3) Line 288 - change to deplete to more accurate word... decrease, impact, diminish, start to deplete, etc.
Authors: Agreed. We have made the appropriate change.
Reviewer: 4) Line 291-292: Seems inaccurate to say that muscle MUST be presented with an optimal pool of EAA precursors to restore the anabolic influence of AA. Based on burns research, the EAAs are not adequate to restore the anabolic response. Therefore, it is really the hormonal amounts/signaling that must be restored and/or enhanced. This alone will (in theory) restore the anabolic effect of protein. I didn't see any evidence provided that EAAs will overcome the anabolic resistance as stated in Comment 1. If such evidence exists, please provide.
Authors: We agree that this is a good point, and used wording that is more appropriate for the information/research we used in the manuscript: hormonal restoration is needed to restore the anabolic response to feeding. Lines: 310 – 311.
Round 2
Reviewer 1 Report
The revised manuscript has been significantly improved.
Reviewer 3 Report
I have no further comments for the authors. There is an outstanding comment with the editor.